# Age and Gender-Specific Pattern of Cardiovascular Disease Risk Factors in Saudi Arabia: A Subgroup Analysis from the Heart Health Promotion Study

**DOI:** 10.3390/healthcare11121737

**Published:** 2023-06-13

**Authors:** Hayfaa Wahabi, Samia Esmaeil, Rasmieh Zeidan, Amr Jamal, Amel A. Fayed

**Affiliations:** 1Research Chair for Evidence-Based Health Care and Knowledge Translation, King Saud University, P.O. Box 800, Riyadh 11421, Saudi Arabia; umlena@yahoo.com (H.W.); sesmaeil@ksu.edu.sa (S.E.); amrjamal@ksu.edu.sa (A.J.); 2Department of Family and Community Medicine, College of Medicine, King Saud University Medical City, P.O. Box 800, Riyadh 11421, Saudi Arabia; 3Cardiac Sciences Department, College of Medicine, King Saud University, P.O. Box 800, Riyadh 11421, Saudi Arabia; ras_zeidan@hotmail.com; 4Clinical Sciences Department, College of Medicine, Princess Nourah bint Abdulrahman University, P.O. Box 84428, Riyadh 11671, Saudi Arabia

**Keywords:** cardiovascular risk, age, gender, Saudi Arabia

## Abstract

Objective: To investigate gender and age-specific distribution patterns of cardiovascular disease risk factors in the Saudi population for tailored health policies. Methods: From the heart health promotion study, 3063 adult Saudis were included in this study. The study cohort was divided into five age groups (less than 40 years, 40–45 years, 46–50 years, 51–55 years and ≥56 years). The prevalence of metabolic, socioeconomic, and cardiac risk was compared between the groups. Anthropometric and biochemical data were gathered using the World Health Organization stepwise approach to chronic disease risk factors. The cardiovascular risk (CVR) was determined using the Framingham Coronary Heart Risk Score. Results: The prevalence of CVR risk increased with age in both genders. Both Saudi men and women exhibit similar propensities for sedentary lifestyles and unhealthy food habits. The prevalence of tobacco smoking was significantly higher and from an early age in males compared to females (28% and 2.7%, respectively, at age 18–29 years). There is no significant difference in either the prevalence of diabetes, hypertension, or metabolic syndrome between men and women before the age of 60 years. Old Saudi females (≥60 years) have a higher prevalence of diabetes (50% vs. 38.7%) and metabolic syndrome (55.9% versus 43.5%). Obesity was more prevalent in females aged 40–49 years onwards (56.2% vs. 34.9% males), with 62.9% of females aged ≥60 years being obese compared to 37.9% of males. Dyslipidaemia prevalence increased with the progression of age, significantly more in males than females. Framingham high-risk scores showed that 30% of males were at high risk of cardiovascular diseases at the age group of 50–59 years, while only 3.7% of the females were considered as such. Conclusions: Both Saudi men and women exhibit similar propensities for sedentary lifestyles and unhealthy food habits, with a marked increase in cardiovascular and metabolic risk factors with age. Gender differences exist in risk factor prevalence, with obesity as the main risk factor in women, while smoking and dyslipidaemia were the main risk factors in men.

## 1. Introduction

Cardiovascular disease (CVD) is a leading cause of morbidity and mortality worldwide, and it is associated with several modifiable risk factors such as hypertension, dyslipidaemia, diabetes mellitus, obesity, smoking, and a sedentary lifestyle [1,2]. Epidemiological studies which used data from the global burden of CVD documented considerable spatiotemporal variation at the levels of exposure to different risk factors and risk-attributable burden [1]. The prevalence of these risk factors differs between men and women [3] and with age; furthermore, risk factors are influenced by various social, cultural, and biological factors; hence, their prevalence differs between countries and regions [1].

It is increasingly recognized that biological differences modify the risk and the onset of developing CVD in women compared to men. Studies confirmed that the onset of CVD in women is, on average, nine years later than in men [4]. This early manifestation of CVD in men may be explained by the high prevalence of some risk factors, which operate from a young age, such as smoking, or the suggested protective effect of oestrogen in pre-menopausal women [5,6]. However, pregnancy and childbirth imposes gender-specific risks for CVD in women. Some pregnancy-related disorders are related to the development of CVD in later life, such as gestational diabetes and preeclampsia [7,8]. In addition, excessive gestational weight gain and postpartum weight retention are associated with the development of many CVD risks [9,10]. Compared to pre-menopausal women, postmenopausal women have a significantly higher prevalence of metabolic syndrome [11] and dyslipidaemia [12,13]. This exponential increase in CVD and its risk factors following menopause may be associated with the lower concentrations of oestrogen and higher concentrations of androgen due to ovarian failure at this stage of the women’s reproductive life [2,14].

Based on the global burden of disease, CVD is the leading cause of premature death in Saudi Arabia, while high body mass index (BMI) is the top risk factor [15].

A recently published systematic review on CVD risk factors in women in Saudi Arabia, which included more than 60 studies, showed a high prevalence of overweight and obesity of 67% and physical inactivity of between 50% to 98% [16]. However, the authors did not conduct analysis based on age, which is imperative to tailor interventions to at-risk age groups.

The results of the PURE study, which included data from Saudi Arabia, showed that men have a greater CVD risk factor burden compared to women and that both CVD incidence and death from CVD are significantly more in men compared to women [17].

Given the increasing burden of CVD in Saudi Arabia, it is important to recognize these differences and to develop age and gender-specific strategies for CVD primordial and primary prevention and management in Saudi Arabia [18]. This may include interventions targeted to address the specific risk factors that are more prevalent in each gender bearing in mind that some risk factors, such as smoking and obesity [19,20], may start at an early age to manifest as CVD later in life.

The aim of this study is to compare the prevalence of CVD risk factors between Saudi men and women to identify gender and age-specific patterns of CVD risk factors in Saudi Arabia.

## 2. Materials and Methods

### 2.1. Consent and Ethics

The study followed the standards of the Helsinki Declaration after receiving approval from King Saud University’s Institutional Review Board (IRB) (reference number 13–3721). All participants signed informed consent forms.

### 2.2. Study Setting

The original cohort from the heart health promotion study included 4500 participants recruited from employee clinics in King Saud University Hospital that serve the employees and their families [21].

### 2.3. Study Population and Sampling Technique

For this study we included a cohort from the heart health promotion study of 3063 Saudi participants. We excluded non-Saudis and pregnant women from this study.

Considering the prevalence of obesity as 25% ± 5% (*p* < 0.01), a power of >0.9 was calculated using STATA/IC14.2.

The study cohort was divided into males and females then each gender group was divided into five age groups (less than 40 years, 40–45 years, 46–50 years, 51–55 years, and ≥56 years). The groups were compared with respect to the prevalence of metabolic and socioeconomic cardiovascular risks (CVR).

### 2.4. Data Collection and Physical Measurements

The sociodemographic data (age, marital status, and educational attainment), data about tobacco use, physical activity, healthy diet, and anthropometric and biochemical measurements were collected using the World Health Organization (WHO) stepwise approach to chronic disease risk factor Surveillance-Instrument v2.1 [22].

All participants were required to fast for at least 12 h before giving blood samples. Glycosylated haemoglobin (HbA1c), high-density lipoprotein cholesterol (HDL-C), low-density lipoprotein cholesterol (LDL-C), total cholesterol (TC), and triglycerides (TG) were measured.

### 2.5. Study Variables

#### 2.5.1. Obesity

Weight and height were measured for all participants. Weight was measured to the nearest 10 g, while height was measured to the nearest 0.1 cm. Body mass index (BMI) was calculated using the formula BMI = weight (kg)/height (m^2^). Based on BMI, the study population was divided into two groups: non-obese (<30 kg/m^2^) and obese (30+ kg/m^2^) [23].

#### 2.5.2. Central Obesity

Waist circumference (WC) was measured in centimetres to the nearest 0.1 cm, using a flexible non-stretchable plastic tape, in a relaxed standing position, during expiration, at the midline between the lower costal margins and the iliac crest parallel to the floor. A WC of 88 cm was used for the diagnosis of central obesity among women, which are cut-off values reported to be applicable to Arab ethnicities [24].

#### 2.5.3. Current Smokers

Smokers were classified as individuals who had smoked at least one cigarette per day for the previous six months, one cigar or water pipe weekly for the last six months, or one waterpipe tobacco smoke/shisha session each month for the prior three months [25].

#### 2.5.4. Physical Inactivity

Participants were deemed physically inactive if they did not meet any of the following WHO standards: 150 min of moderate activity each week or 60 min of vigorous activity [26].

#### 2.5.5. Low Fruit and Vegetable Intake

According to the WHO, any subject who had less than five servings (400 gm) of fruit and/or vegetables per day was considered as having inadequate intake [27,28].

#### 2.5.6. Hypertension

Both systolic and diastolic pressures were measured at two readings set five minutes apart; the average of the two readings was used. Hypertension was defined as being previously diagnosed as hypertensive and currently using any anti-hypertensive medications or having high blood pressure readings according to the Seventh Report of the Joint National Committee on Prevention, Detection, Evaluation, and Treatment of High Blood Pressure (JNC7) [29].

#### 2.5.7. Diabetes Mellitus

Diabetes mellitus was defined as per WHO and American Diabetes Association criteria or by the subject reporting of being previously diagnosed as diabetic and using anti-diabetes medication [30].

#### 2.5.8. Cardiovascular Risk (CVR) Scores

Scores were calculated for all participants using the Framingham Coronary Heart Risk Score (FRS), which is one of the most extensively used cardiovascular risk calculators in clinical practice. It was used to calculate the 10-year risk of coronary heart disease, where the cohort was subdivided according to their scores into three categories: low-risk score (<10%), intermediate (10–20%), and high (>20%) [31].

#### 2.5.9. Metabolic Syndrome (MetS)

If participants satisfied at least three of the five criteria listed in the Third Report of the National Cholesterol Education Program (NCEP) Adult Treatment Panel III) (NCEP-ATPIII) criteria, they were considered to have metabolic syndrome [32].

#### 2.5.10. Dyslipidaemia

Dyslipidaemia was considered according to definitions adopted by the National Cholesterol Education Program F(NCEP) criteria for dyslipidaemia (elevated cholesterol, elevated TG, high HDL-C level, and low LDL-C).

### 2.6. Statistical Analysis

Continuous variables were reported as means with standard deviations. Categorical variables were presented as frequencies with equivalent percentages, and Pearson’s chi-square test was used for the comparison of different proportions.

## 3. Results

The sociodemographic characteristics of the participants are shown in Table 1. While more than 50% of the female participants of the age group 60 years and above were illiterate, only 3.2% of males in this group were illiterate. However, this high rate of illiteracy was not observed in young age groups, nor were the marked differences between females and males in different levels of education (Table 1). Most of the participants were married by the age of 30–39 years; however, by the age of ≥60 years, significantly more women were widowed than men (Table 1). While 23–36% of the male participants smoke tobacco, only 1.4–3.0% of the female participants do so; however, female participants were significantly less physically active compared to the male participants in all age groups (Table 1). Nevertheless, 90–80% of the participants have poor dietary habits (Table 1, Figure 1).

The prevalence of cardiovascular risk factors progressively increased with the increase in age in both males and females. The prevalence of BMI 30+ kg/m^2^ was 25.4% among males of the age group 18–29 years, yet it was 16.5% among females of the same age group. This difference in the prevalence of obesity was less obvious in the age group 30–39 years; we noticed that in the age group of 40–49 years, the prevalence of obesity in women (56.2%) progressively increased to supersede the prevalence in men (34.9%) in the same age group. At the age of ≥60 years, 62.9% of females were obese, while only 37.9% of males were obese (Table 2, Figure 2).

We noticed the same pattern of distribution of the prevalence of central obesity between females and males in different age groups, where females have a lower prevalence at the age group of 18–29 years compared to males (12% vs. 7.9%), then at the age of 40–49 years the prevalence is higher in females compared to men (37.8% vs. 22.9%), and at the age ≥60 years prevalence in females was double that in the males (51.7% vs. 24.2%) (Table 2, Figure 3).

The prevalence of diabetes in both males and females was low until the age of 49 years (males 1.2–16% and females 1.9–16.5%). Then, we noticed a progressive increase in the prevalence in both genders in the age group of 50–59 years, where the prevalence in males was slightly more than in the females (35.3% vs. 30.4%); however, in the age group ≥60 years, almost 50% of the females were diabetic, while 38.7% of the males were diabetic (Table 2, Figure 4).

There was a progressive increase in the prevalence of hypertension in both genders with the increase in age. The prevalence was low in the age groups 18–29 and 30–39 years (males 6.5–13.8% and females 2.4–5.4%). In the age group 50–59 years, 40% of males and females had hypertension, and in the age group of ≥60 years, 62.1% and 67.8% of males and females, respectively, had hypertension (Table 2, Figure 5).

The prevalence of all forms of dyslipidaemia was aggravated with the progression of age among both males and females. In addition, metabolic syndrome was initially more prevalent in women than in men at earlier ages (3% vs. 7.7%), but subsequently, it surpassed that of men at older ages (55.9% against 43.5%) (Table 2, Figure 6, Figure 7, Figure 8 and Figure 9).

Calculation of Framingham risk for males and females showed that 30% of males were at high risk of CVD in the age group 50–59 years, while only 3.7% of females were considered as such. In the age group ≥60 years, 67.7% of males were at high risk for CVD, while 28.7% of the females were in the high-risk group (Table 2, Figure 10).

## 4. Discussion

This study showed that both Saudi men and women have a similar propensity to sedentary lifestyles and unhealthy food habits and that more than one quarter of the male participants were tobacco smokers. In addition, there was a significant increase in the prevalence of cardiovascular and metabolic risk factors with age, both in men and women. However, there are important gender differences in the prevalence of those risk factors, with women generally having a lower prevalence of risk factors among young age groups but catching up and sometimes surpassing men in old age groups.

The positive role of a plant-based diet in the prevention of CVD has been repeatedly recognized [33,34]. Such a diet was found to lower dyslipidaemia, one of the main modifiable risk factors of CVD [35]. The adoption of an energy-balanced, plant-based diet at a national level was found to be associated with up to 22% reduction in premature mortality [36]. Based on the results of this study that showed a low prevalence of fruits and vegetable consumption (Table 1), such a strategy is urgently needed in Saudi Arabia to improve consumption and reduce premature death [37].

The high prevalence of physical inactivity in both genders, nevertheless, significantly more in women in all age groups observed in this study, is consistent with previous reports from Saudi Arabia (Table 1) [38,39]. In addition, the high prevalence of inactivity among women and its association with CVD has been reported in previous studies as a global modifiable risk factor for CVD in women [40]. The positive impact of physical activity in the prevention and treatment of CVD is well-proven [41]. The results of the INTERHEART study, which investigated modifiable risk factors for CVD and analyzed data from 52 countries, showed that regular physical activity is associated with reduced odds of developing CVD (OR, 0.86, CI 0.76–0.97), with more than 12% population attributable risk (PAR) [2].

Another noteworthy socioeconomic determinant of health in this study is the significant difference in the level of education between men and women, starting in the age group ≥ 40 years, with obvious differences in education in the age group ≥60 years, where 50% of women were illiterate compared to 3.3% of men (Table 1).

Many studies confirmed the association between low levels of education and CVD and mortality [42,43]. A low level of education is associated with poor health literacy, such as nonadherence to medication, and risky health behavior, such as physical inactivity and tobacco smoking [44,45]. However, the extent of the effect of education as a risk factor for CVD was not quite clear in this study. Some risky behaviors, such as smoking, are significantly more prevalent among men, who are more educated, than among women; in addition, sedentary lifestyles and unhealthy food habits are common in both genders (Table 1).

The findings of this study revealed that the prevalence of metabolic syndrome, including abdominal obesity, hypertension, diabetes, and dyslipidaemia, which are the main risk factors for CVD, is initially lower in women compared to men at younger ages, but then it exceeds that of men at older ages (Table 2, Figure 2, Figure 3, Figure 4, Figure 5, Figure 6, Figure 7, Figure 8 and Figure 9). This finding is consistent with other studies that have shown a higher prevalence of metabolic syndrome in postmenopausal women compared to men of the same age group [11,46]. This difference in the prevalence may be associated with hormonal changes, including oestrogen deficiency and high androgens levels that occur during this stage of women’s life [11]. Furthermore, in this study, women had a higher prevalence of general obesity as indicated by BMI 30+ kg/m^2^, which is almost double the prevalence in men from the age of 40 years and more (Table 2, Figure 1). This high prevalence of obesity in Saudi women compared to men in this age group may be due to excessive gestational weight gain and postpartum weight retention [9,10,47] during the reproductive stage of their life, as most of these women are married and have completed their families (Table 1). On the other hand, the high obesity prevalence noted in older women may be explained by the hormonal effect of menopause transition and menopause, as reported in earlier studies [48,49].

The Framingham high-risk category for developing CVD was less frequent in women compared to men in this study (Table 2 and Figure 6), an observation consistent with previous studies which proved that CVD manifests nine years earlier in men compared to women [2,4,50]. The earlier manifestation of CVD in men may be explained by the earlier exposure of men to the main risk factors for such disease [51], as is apparent in this study. An additional important difference between men and women in the risk of CVD in this study is the high rate of tobacco smoking among men compared to women (Table 1). Tobacco smoking was found to increase the odds of myocardial infarction by nearly three-fold; in addition, it has the highest PAR of 35.7% compared to the other known risk factors such as abdominal obesity (PAR 20%), hypertension (PAR 17.9%), and diabetes (PAR 9.9%) [2]. Another significant difference between men’s and women’s risk of CVD is the higher prevalence of abnormal lipid profiles in men, throughout the age groups, compared to the women, as seen in Table 2. These results are consistent with the findings of other studies, which showed dyslipidaemia prevalence higher in men than in women [52]. In addition, a recently published report showed an increased trend of disability and premature death in Saudi Arabia during the recent decades from dyslipidaemia, while in most of the other countries, such a trend has been static [53]. Dyslipidaemia is associated with atherosclerotic cardiovascular disease [54]. Globally, smoking and dyslipidaemia are the two most important risk factors for CVD [2], responsible for two-thirds of PAR of myocardial infarction [2].

To reduce the burden of CVD in Saudi Arabia, health policy should be tailored to address the modifiable risk factors with the greatest bearing on the group of the community most affected by those factors, yet not ignoring the other globally operating risk factors [18,55]. The results of this study showed that tobacco smoking and dyslipidaemia are the main risk factors in Saudi men, especially the former, which is prevalent in the young age group. Tobacco smoking cessation intervention may be challenging [56]; nevertheless, scaling up such interventions and using a multifaceted approach will have an impact on reducing and preventing the use of tobacco among school children and college students [57]. Screening and management of dyslipidaemia have long been advocated to reduce the burden of CVD in the Middle East. There is great emphasis on educating healthcare providers and the community on the importance of treating dyslipidaemia and on establishing surveillance systems for CVD to monitor the effects of interventions [58].

It is well recognized that obesity worsens other CVD risk factors by its augmenting effects on dyslipidaemia, high blood pressure, peripheral insulin resistance, and inflammation [59,60]. Based on the results of this study and previous reports from Saudi Arabia [61,62], which showed a distinctive pattern of obesity in Saudi women, obesity management and prevention in women should be focused on weight management around pregnancy and menopause.

### 4.1. Strength and Limitations

This study provided important information about the modifiable risk factors and at-risk groups based on gender and age in the Saudi community. This information is imperative for strategically targeting at-risk populations with sustainable and cost-effective interventions as part of intermediate and long-term strategies to reduce the burden of CVD in Saudi Arabia [63]. We are aware of the limitations of this study, including the cross-sectional nature of this sub-group analysis which did not include follow-up of participants. Such follow-up is important to explore the effects of other non-modifiable risks, such as genetics and their influence on modifiable risk factors [64].

### 4.2. Implication to Practice

The Kingdom of Saudi Arabia, as a high-income country, is in a good position to implement primordial and primary prevention of CVD, which is the leading cause of mortality and morbidity in the country. Addressing social determinants of health such as maternal and child nutrition, smoking cessation, advocating healthy food habits, and improving physical activity should be the main component of any strategy to address CVD risk management [18]. In addition to the known primary, secondary, and tertiary prevention programs [18], this study suggested that special attention should be paid to the following groups of the community:Preventive interventions should target school children and university students to stop tobacco smoking initiation and to implement active programs for tobacco smoking cessation.Programs for screening and treatment of dyslipidaemia, especially for men, should be an integral part of any preventive program for CVD.Obesity prevention and management for women should be targeted at the reproductive age group to prevent excessive weight gain during pregnancy and postpartum weight retention.Programs for primary and secondary prevention of CVD in old men and women should include the prevention and management of obesity in postmenopausal women.Saudi men and women should be at the center of preventive and curative care for CVD.

### 4.3. Implication to Research

Extensive research should aim at investigating the effective means of improving the physical activity of individuals in the Saudi community as part of lifestyle modification.Research should focus on effective and acceptable means of changing the type and habit of food consumption in the community and among at-risk populations [65,66].Surveillance systems for CVD and all the risk factors should be established to monitor the effectiveness of interventions.

## 5. Conclusions

This study showed that all cardiometabolic risk factors are prevalent in both men and women; however, they show higher prevalence at earlier ages in men compared to women. A small proportion of Saudi men and women assume an active lifestyle and adopt a plant-based diet. The main risk factors for men are tobacco smoking and dyslipidaemia, while obesity is the main risk in females.

## Figures and Tables

**Figure 1 healthcare-11-01737-f001:**
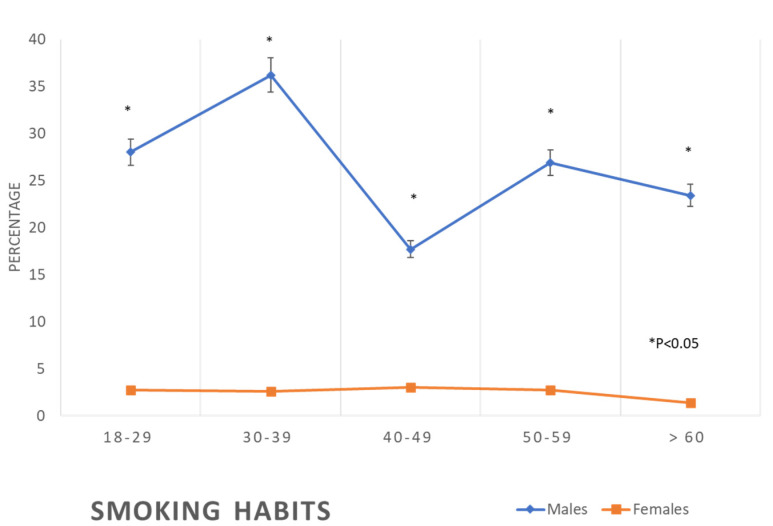
Prevalence of smoking in the studied sample according to age and gender (*: *p* < 0.05).

**Figure 2 healthcare-11-01737-f002:**
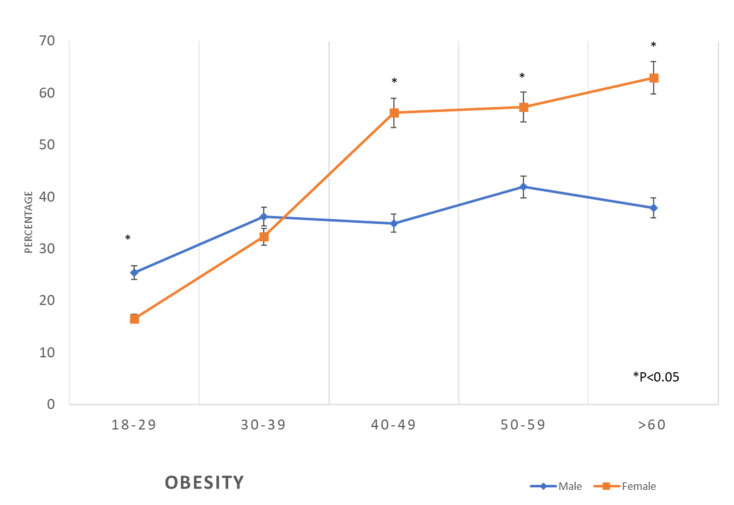
Prevalence of obesity in the studied sample according to age and gender (*: *p* < 0.05).

**Figure 3 healthcare-11-01737-f003:**
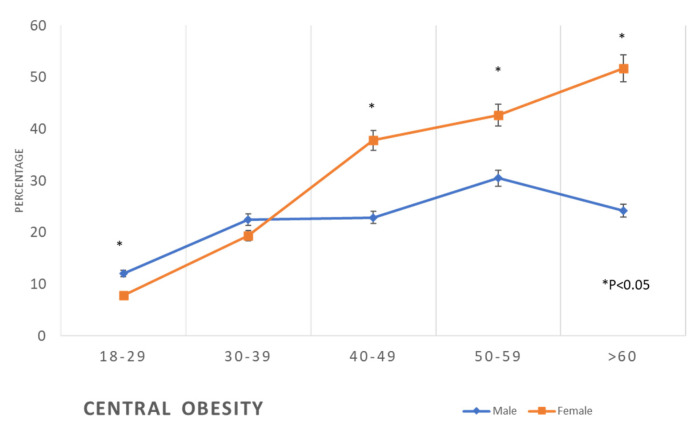
Prevalence of central obesity in the studied sample according to age and gender (*: *p* < 0.05).

**Figure 4 healthcare-11-01737-f004:**
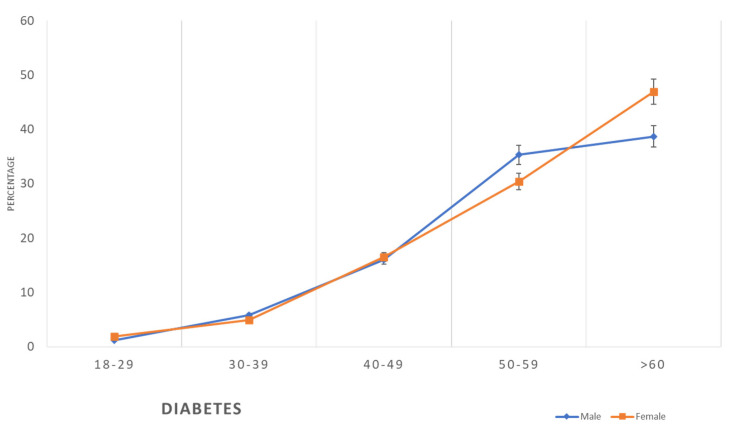
Prevalence of diabetes in the studied sample according to age and gender.

**Figure 5 healthcare-11-01737-f005:**
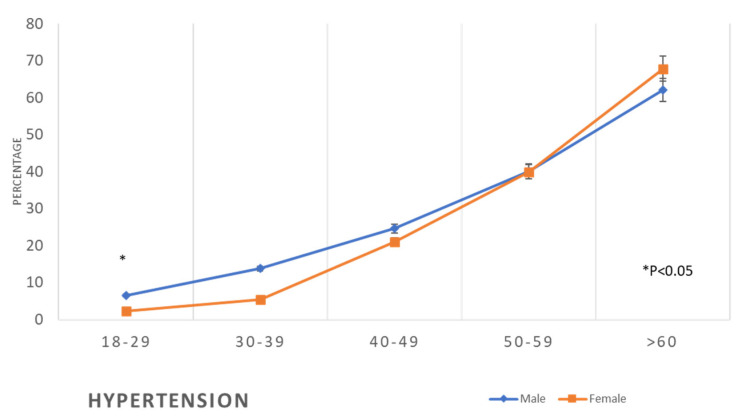
Prevalence of hypertension in the studied sample according to age and gender(*: *p* < 0.05).

**Figure 6 healthcare-11-01737-f006:**
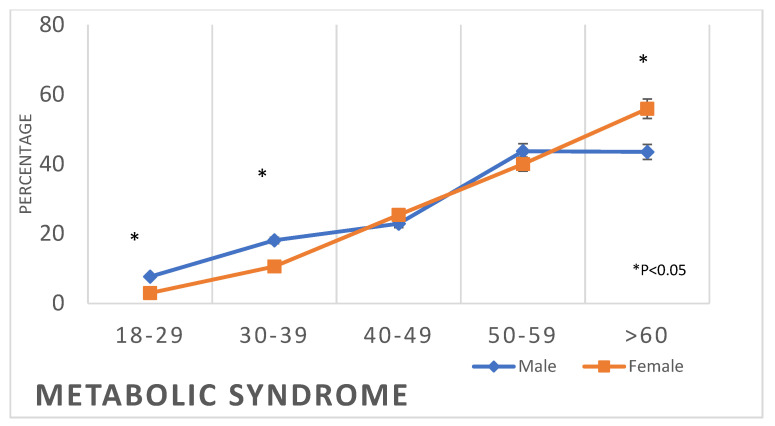
Prevalence of metabolic syndrome in the studied sample according to age and gender.

**Figure 7 healthcare-11-01737-f007:**
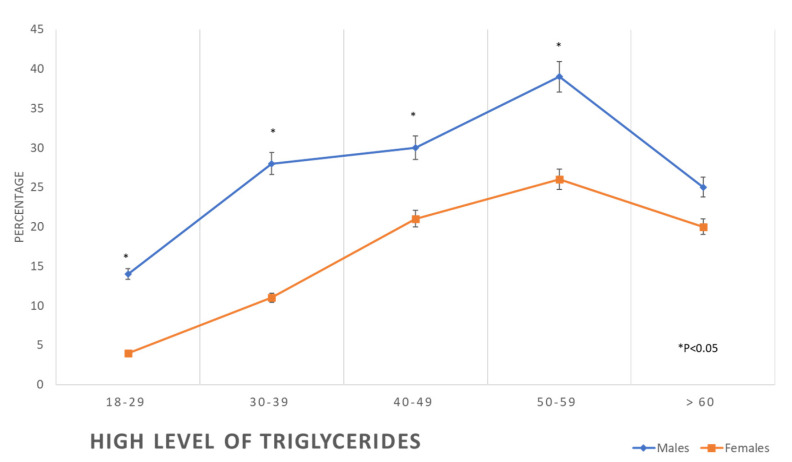
Prevalence of high levels of triglycerides in the studied sample according to age and gender.

**Figure 8 healthcare-11-01737-f008:**
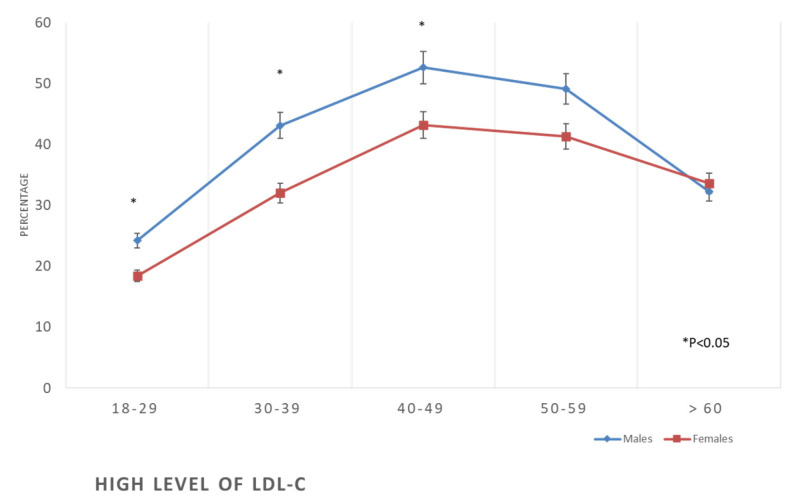
Prevalence of high levels of low-density lipoproteins in the studied sample according to age and gender.

**Figure 9 healthcare-11-01737-f009:**
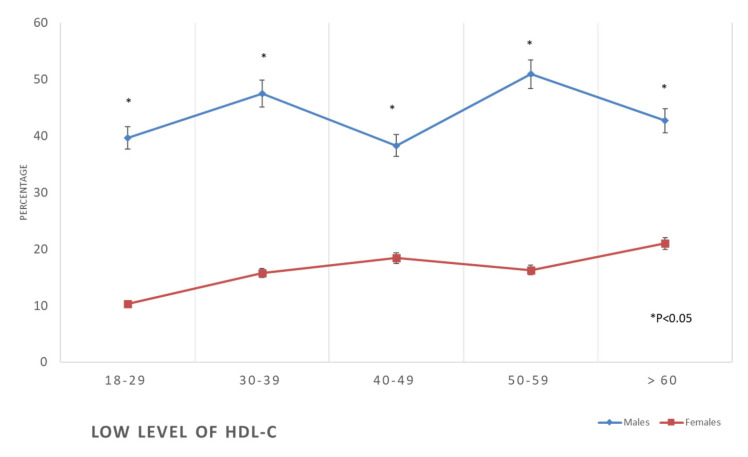
Prevalence of low levels of high-density lipoproteins in the studied sample according to age and gender.

**Figure 10 healthcare-11-01737-f010:**
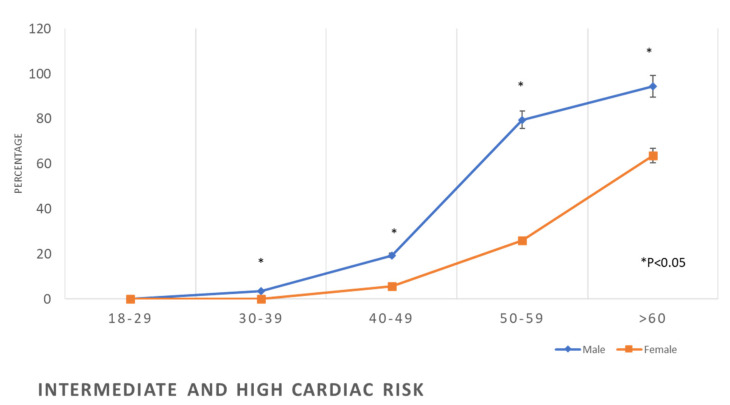
Prevalence of intermediate and high cardiac risk in the studied sample according to age and gender.

**Table 1 healthcare-11-01737-t001:** Distribution of the sociodemographic characteristics of the studied sample according to age and gender.

		Ages Groups																			
		18–29 Years Old				30–39 Years Old				40–49 Years Old				50–59 Years Old				>60 Years Old			
		Gender				Gender				Gender				Gender				Gender			
		Male		Female		Male		Female		Male		Female		Male		Female		Male		Female	
		N	%	N	%	N	%	N	%	N	%	N	%	N	%	N	%	N	%	N	%
Education levels	University	225	(54.3)	350	(55.4)	234	(84.8)	297	(76.7)	147	(84.0)	203	(54.9)	136	(81.4)	157	(41.9)	91	(73.4)	30	(21.0)
	school	189	(45.7)	276	(43.7)	42	(15.2)	86	(22.2)	27	(15.4)	147	(39.7)	30	(18.0)	168	(44.8)	29	(23.4)	41	(28.7)
	Illiterate	0	(0.0)	6	(0.9)	0	(0.0)	4	(1.0)	1	(0.6)	20	(5.4)	1	(0.6)	50	(13.3)	4	(3.2)	72	(50.3)
	*p*-value	0.12				0.02				<0.01				<0.01				<0.01			
Marital status	Married	163	(39.4)	211	(33.4)	259	(93.8)	293	(75.7)	170	(97.1)	339	(91.6)	167	(100.0)	347	(92.5)	123	(99.2)	115	(80.4)
	Single	250	(60.4)	404	(63.9)	16	(5.8)	70	(18.1)	2	(1.1)	10	(2.7)	0	(0.0)	2	(0.5)	0	(0.0)	3	(2.1)
	Widowed and divorced	1	(0.2)	17	(2.7)	1	(0.4)	24	(6.2)	3	(1.7)	21	(5.7)	0	(0.0)	26	(6.9)	1	(0.8)	25	(17.5)
	*p*-value	<0.01				<0.01				0.05				<0.01				<0.01			
Smoking	No	298	(72.0)	615	(97.3)	176	(63.8)	377	(97.4)	144	(82.3)	359	(97.0)	122	(73.1)	365	(97.3)	95	(76.6)	141	(98.6)
	Yes	116	(28.0)	17	(2.7)	100	(36.2)	10	(2.6)	31	(17.7)	11	(3.0)	45	(26.9)	10	(2.7)	29	(23.4)	2	(1.4)
	*p*-value	<0.01				<0.01				<0.01				<0.01				<0.01			
Dietary Habits	Poor	375	(90.6)	583	(92.2)	233	(84.4)	356	(92.0)	142	(81.1)	311	(84.1)	137	(82.0)	328	(87.5)	98	(79.0)	120	(83.9)
	Good	39	(9.4)	49	(7.8)	43	(15.6)	31	(8.0)	33	(18.9)	59	(15.9)	30	(18.0)	47	(12.5)	26	(21.0)	23	(16.1)
	*p*-value	0.34				<0.01				0.39				0.09				0.30			
Physical activity	Inactive	240	(58.0)	517	(81.8)	177	(64.1)	340	(87.9)	125	(71.4)	334	(90.3)	117	(70.1)	344	(91.7)	91	(73.4)	137	(95.8)
	Active	174	(42.0)	115	(18.2)	99	(35.9)	47	(12.1)	50	(28.6)	36	(9.7)	50	(29.9)	31	(8.3)	33	(26.6)	6	(4.2)
	*p*-value	<0.01				<0.01				<0.01				<0.01				<0.01			

**Table 2 healthcare-11-01737-t002:** Cardiovascular risk profile of the studied sample according to age and gender.

		Ages Groups																			
		18–29 Years Old				30–39 Years Old				40–49 Years Old				50–59 Years Old				>60 Years Old			
		Gender				Gender				Gender				Gender				Gender			
		Male		Female		Male		Female		Male		Female		Male		Female		Male		Female	
		N	%	N	%	N	%	N	%	N	%	N	%	N	%	N	%	N	%	N	%
DM	Normal	409	(98.8)	620	(98.1)	260	(94.2)	368	(95.1)	147	(84.0)	309	(83.5)	108	(64.7)	261	(69.6)	76	(61.3)	76	(53.1)
	Diabetes	5	(1.2)	12	(1.9)	16	(5.8)	19	(4.9)	28	(16.0)	61	(16.5)	59	(35.3)	114	(30.4)	48	(38.7)	67	(46.9)
	*p*-value	0.39				0.61				0.89				0.26				0.18			
HTN	Normal	387	(93.5)	617	(97.6)	238	(86.2)	366	(94.6)	132	(75.4)	292	(78.9)	100	(59.9)	225	(60.0)	47	(37.9)	46	(32.2)
	HTN	27	(6.5)	15	(2.4)	38	(13.8)	21	(5.4)	43	(24.6)	78	(21.1)	67	(40.1)	150	(40.0)	77	(62.1)	97	(67.8)
	*p*-value	<0.01				<0.01				0.36				0.98				0.33			
Obesity	BMI < 30	309	(74.6)	528	(83.5)	176	(63.8)	262	(67.7)	114	(65.1)	162	(43.8)	97	(58.1)	160	(42.7)	77	(62.1)	53	(37.1)
	BMI ≥30	105	(25.4)	104	(16.5)	100	(36.2)	125	(32.3)	61	(34.9)	208	(56.2)	70	(41.9)	215	(57.3)	47	(37.9)	90	(62.9)
	*p*-value	<0.01				0.29				<0.01				<0.01				<0.01			
Central Obesity	No	364	(87.9)	582	(92.1)	214	(77.5)	312	(80.6)	135	(77.1)	230	(62.2)	116	(69.5)	215	(57.3)	94	(75.8)	69	(48.3)
	Yes	50	(12.1)	50	(7.9)	62	(22.5)	75	(19.4)	40	(22.9)	140	(37.8)	51	(30.5)	160	(42.7)	30	(24.2)	74	(51.7)
	*p*-value	0.03				0.33				<0.01				<0.01				<0.01			
TG	normal	357	(86.2)	607	(96.0)	199	(72.1)	344	(88.9)	122	(69.7)	294	(79.5)	102	(61.1)	277	(73.9)	93	(75.0)	115	(80.4)
	High TG	57	(13.8)	25	(4.0)	77	(27.9)	43	(11.1)	53	(30.3)	76	(20.5)	65	(38.9)	98	(26.1)	31	(25.0)	28	(19.6)
	*p*-value	<0.01				<0.01				0.01				<0.01				0.29			
TC/HDL-c ratio	<5	320	(77.3)	602	(95.3)	165	(59.8)	345	(89.1)	100	(57.1)	305	(82.4)	95	(56.9)	304	(81.1)	92	(74.2)	125	(87.4)
	5+	94	(22.7)	30	(4.7)	111	(40.2)	42	(10.9)	75	(42.9)	65	(17.6)	72	(43.1)	71	(18.9)	32	(25.8)	18	(12.6)
	*p*-value	<0.01				<0.01				<0.01				<0.01				<0.01			
HDL-c	Normal	250	(60.4)	567	(89.7)	145	(52.5)	326	(84.2)	108	(61.7)	302	(81.6)	82	(49.1)	314	(83.7)	71	(57.3)	113	(79.0)
	Low HDL	164	(39.6)	65	(10.3)	131	(47.5)	61	(15.8)	67	(38.3)	68	(18.4)	85	(50.9)	61	(16.3)	53	(42.7)	30	(21.0)
	*p*-value	<0.01				<0.01				<0.01				<0.01				<0.01			
LDL-c	Normal	314	(75.8)	516	(81.6)	157	(56.9)	263	(68.0)	83	(47.4)	210	(56.8)	85	(50.9)	220	(58.7)	84	(67.7)	95	(66.4)
	High LDL	100	(24.2)	116	(18.4)	119	(43.1)	124	(32.0)	92	(52.6)	160	(43.2)	82	(49.1)	155	(41.3)	40	(32.3)	48	(33.6)
	*p*-value	0.02				<0.01				0.04				0.09				0.80			
Total Cholesterol	Normal	331	(80.0)	483	(76.4)	168	(60.9)	246	(63.6)	80	(45.7)	185	(50.0)	79	(47.3)	191	(50.9)	84	(67.7)	89	(62.2)
	High TC	83	(20.0)	149	(23.6)	108	(39.1)	141	(36.4)	95	(54.3)	185	(50.0)	88	(52.7)	184	(49.1)	40	(32.3)	54	(37.8)
	*p*-value	0.18				0.48				0.35				0.44				0.35			
MetS	No	382	(92.3)	613	(97.0)	226	(81.9)	346	(89.4)	135	(77.1)	276	(74.6)	94	(56.3)	225	(60.0)	70	(56.5)	63	(44.1)
	Yes	32	(7.7)	19	(3.0)	50	(18.1)	41	(10.6)	40	(22.9)	94	(25.4)	73	(43.7)	150	(40.0)	54	(43.5)	80	(55.9)
	*p*-value	<0.01				<0.01				0.52				0.42				0.04			
FRS	Low risk	414	(100.0)	632	(100.0)	266	(96.4)	387	(100.0)	141	(80.6)	349	(94.3)	34	(20.4)	277	(73.9)	7	(5.6)	52	(36.4)
	Intermediate risk	0	(0.0)	0	(0.0)	9	(3.3)	0	(0.0)	31	(17.7)	20	(5.4)	82	(49.1)	84	(22.4)	33	(26.6)	50	(35.0)
	High risk	0	(0.0)	0	(0.0)	1	(0.4)	0	(0.0)	3	(1.7)	1	(0.3)	51	(30.5)	14	(3.7)	84	(67.7)	41	(28.7)
	*p*-value	.				<0.01				<0.01				<0.01				<0.01			

DM: Diabetes Mellites, HTN: Hypertension, TG: Triglycerides, TC/HDL-c ratio: total cholesterol/high-density lipoproteins ratio, HDL-c: High-density lipoprotein cholesterol, LDL-c: Low-density lipoprotein cholesterol, MetS: Metabolic Syndrome, FRS: Framingham’s cardiac risk score.

## Data Availability

Most of the data needed are included in the published article. However, more data are available from the King Saud University Ethics Committee for researchers for those who meet the criteria for access to confidential data. The ethics committee contact details for data requests are: irb@ksu.edu.sa. This contact point is completely independent of all researchers.

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
