# Peer review of "Age and Gender-Specific Pattern of Cardiovascular Disease Risk Factors in Saudi Arabia: A Subgroup Analysis from the Heart Health Promotion Study"

_healthcare, 2023, doi:10.3390/healthcare11121737_

Round 1

Reviewer 1 Report

In this manuscript, Hayfaa Wahabi and other authors demonstrated that there is a significant increase in the prevalence of cardiovascular and metabolic risk factors with age both in men and women. They also claimed that gender differences exist in risk factor prevalence, with obesity as the main risk factor in women while smoking and dyslipidaemia were the main risk factors in men. The main context is well-organized, and the experiment design is overall reasonable. But the results do not strongly support all the conclusions and there are some minor issues need to be addressed.

Firstly, you claimed that obesity was the main risk factor in women but from table 2, figure 1 and figure 2 we can only get the information that in some age group women have high obesity ratio but there is no evidence showing obesity is the cause.

Second, you also claimed that smoking and dyslipidaemia were the main risk factor in men, but from the figures you demonstrated we can not get the conclusion that these factors are the causes.

Lastly, in figure 1-6, whether the p values of each comparison is significant or not should be labelled in the graphs.

Besides, in line 251 the table title should be Table 1 instead of Table 2.

Author Response

We would like to thank the reviewers for their constructive comments and below is point by point reply to their comments

Reviewer 1

Firstly, you claimed that obesity was the main risk factor in women but from table 2, figure 1 and figure 2 we can only get the information that in some age group women have high obesity ratio but there is no evidence showing obesity is the cause.

Obesity has been proven as a risk factor for CVD, as well as it precipitates other risk factors such as diabetes and hypertension (1,2,3). As the objective of this study was to compare the risk factors of CVD between men and women at multiple points of age in their lives, we found that from the age of 40 years and above women have a significantly high prevalence of obesity and central obesity compared to men as seen in figure 1 &2  

Second, you also claimed that smoking and dyslipidaemia were the main risk factor in men, but from the figures you demonstrated we cannot get the conclusion that these factors are the causes.

The conclusion about smoking and dyslipidaemia was based on the significantly high prevalence of these factors in men compared to women (please see Table 2, line 178, TG, HDL, and LDL-C). men compared to women.

As for smoking please see Table 1 where smoking is compared between men and women with significantly high prevalence in men all through the age groups.

We have added 4 figures to demonstrate dyslipidaemia and smoking prevalence

Lastly, in figure 1-6, whether the p values of each comparison are significant or not should be labelled in the graphs.

Besides, in line 251 the table title should be Table 1 instead of Table 2.

Significant P-vales were labeled in all figures and the insignificant P-values are added in the tables at each variable.

Obesity and central obesity prevalence were reported in Table 2

Reviewer 2 Report

I think the manuscript is well written. But it can be improved. Can authors make correlations between factors other than age and gender?

Please state the p-value for each figure.

Please show the figures with a confidence interval

Author Response

Reviewer 2

Can authors make correlations between factors other than age and gender?

In the heart health promotion study, many subgroups were studied to investigate different groups with different risk factors (4), however, this study has the main objective to investigate the difference in the distribution of the risk factors between men and women to advise health policy about the population at specific risk to focus interventions with the objective to reduce CVD prevalence.  

Please state the p-value for each figure.

Done

Please show the figures with a confidence interval

Done

Reviewer 3 Report

This article reports on a cohort study about the age and gender-specific pattern of CVD risk factors in Saudi Arabia, and the database was from the heart health promotion study, and subgroup analysis was conducted. The result showed gender differences in risk factor prevalence, and increasing CV and metabolic risk factors with age in both gender.

Specific comments

1.     About one of the study variables, low fruit and vegetable intake, could you explain how the intake be estimated? Maybe you could provide more information in this part, and how about the reliability of the data because I think the amount of intake is difficult to check.

2.     Could I say that the population in this study is in higher socioeconomic level in Saudi Arabia? (employees and their families in King Saud University Hospital?) If so, could the result in this study be representative for the whole country?

3.     I noticed that the education level in age group > 40 y/o revealing large difference between male and female. Is this difference due to previous government policy? If so, could this difference contribute to your study result? Or could you discuss the possible effect from the education level or government policy   

Author Response

Reviewer 3

1.     About one of the study variables, low fruit and vegetable intake, could you explain how the intake be estimated? Maybe you could provide more information in this part, and how about the reliability of the data because I think the amount of intake is difficult to check.

Insufficient consumption of fruits and vegetables was defined as when an individual fails to meet the recommended daily intake of at least five servings (equivalent to 400gm) as suggested by the World Health Organization (WHO)

Different questions about the average number of daily servings of vegetables and fruits were included in the self-administered questionnaire, categorizations of responses based on the WHO guidelines were created then the classification of participants into two groups of adequate and inadequate intake was used as the study variable.

2.     Could I say that the population in this study is in higher socioeconomic level in Saudi Arabia? (employees and their families in King Saud University Hospital?) If so, could the result in this study be representative for the whole country?

The study population included all the hospital staff and their families including university clinical professors, junior doctors, lab technicians, nurses … even genitors and security personnel, hence we believe it can represent the Saudi community which is generally high to middle-income.

3.    I noticed that the education level in age group > 40 y/o revealing large difference between male and female. Is this difference due to previous government policy? If so, could this difference contribute to your study result? Or could you discuss the possible effect from the education level or government policy   

We believe this difference is not due to government policy but to certain beliefs and customs in the community during the previous century, because girls’ education in Saudi Arabia started in 1956 (5).

We agree with the review about the significant difference in education and its importance, hence we added a paragraph to the discussion (highlighted in yellow)

We believe the influence of education on the distribution of CVD in this study should be approached with caution, as it affects the elderly sector of the community with no reflection on risky behavior or lifestyle. The consumption of fruit and vegetables and physical activity is low irrespective of age, gender, or education.  While smoking is significantly high in all age groups of men, who are more educated, than in women

Round 2

Reviewer 3 Report

 This article reports on a research to show the age- and gender-specific pattern of CVD risk factors in Saudi Arabia, and I think the result reflect population with higher socioeconomic status. 

The author has modified according to the requirements, I have no more questions.